# Synthetic mRNAs Containing Minimalistic Untranslated Regions Are Highly Functional In Vitro and In Vivo

**DOI:** 10.3390/cells13151242

**Published:** 2024-07-24

**Authors:** Shahab Mamaghani, Rocco Roberto Penna, Julia Frei, Conrad Wyss, Mark Mellett, Thomas Look, Tobias Weiss, Emmanuella Guenova, Thomas M. Kündig, Severin Lauchli, Steve Pascolo

**Affiliations:** 1Department of Dermatology, University Hospital Zurich (USZ), University of Zurich (UZH), Raemistrasse 100, 8091 Zurich, Switzerland; shahab.abdolnejadmamaghani@usz.ch (S.M.); rocco.penna@usz.ch (R.R.P.); julia.frei@usz.ch (J.F.); conrad.wyss@usz.ch (C.W.); mark.mellett@usz.ch (M.M.); thomas.look@usz.ch (T.L.); thomas.kuendig@usz.ch (T.M.K.); severin.laeuchli@stadtspital.ch (S.L.); 2Faculty of Science, University of Zurich, 8006 Zurich, Switzerland; 3Faculty of Medicine, University of Zurich, 8006 Zurich, Switzerland; 4Department of Neurology, Clinical Neuroscience Center, University Hospital Zurich (USZ), University of Zurich (UZH), Frauenklinikstrasse 26, 8091 Zurich, Switzerland; tobias.weiss@usz.ch; 5Lausanne University Hospital (CHUV), University of Lausanne, 1015 Lausanne, Switzerland; emmanuella.guenova@unil.ch

**Keywords:** mRNA, in vitro transcription, untranslated regions, ivt mRNA, UTR

## Abstract

Synthetic mRNA produced by in vitro transcription (ivt mRNA) is the active pharmaceutical ingredient of approved anti-COVID-19 vaccines and of many drugs under development. Such synthetic mRNA typically contains several hundred bases of non-coding “untranslated” regions (UTRs) that are involved in the stabilization and translation of the mRNA. However, UTRs are often complex structures, which may complicate the entire production process. To eliminate this obstacle, we managed to reduce the total amount of nucleotides in the UTRs to only four bases. In this way, we generate minimal ivt mRNA (“minRNA”), which is less complex than the usual optimized ivt mRNAs that are contained, for example, in approved vaccines. We have compared the efficacy of minRNA to common augmented mRNAs (with UTRs of globin genes or those included in licensed vaccines) in vivo and in vitro and could demonstrate equivalent functionalities. Our minimal mRNA design will facilitate the further development and implementation of ivt mRNA-based vaccines and therapies.

## 1. Introduction

The primary structure of messenger RNA typically consists of five elements (from 5′ to 3′) [1]: a cap structure, a 5′ untranslated region (UTR) of a few dozen nucleotides between the cap and the start codon, a coding sequence, (from a start to a stop codon), a 3′ UTR that is usually a few hundred nucleotides following the stop codon, and the poly-A tail. UTRs are involved in the stability and translation efficacy of the mRNA. In the early 1980s, thanks to the use of purified RNA polymerase [2] and then of recombinant RNA polymerase [3], synthetic mRNA could be produced in vitro. The first report on the functionality of synthetic mRNA from 1984 used mRNA coding for the xenopus beta globin protein [4]. On top of the coding sequence, this ivt mRNA contained the natural 5′ and 3′ UTRs of the xenopus beta globin gene and a poly-A(23) C(30) tail. Then, the authors replaced the coding sequence of the globin with the sequence of the human beta interferon, keeping the xenopus globin 5′ and 3′ UTRs, and could show that the chimeric synthetic mRNA was functional (producing human interferon when injected in frog oocytes and used in a wheat germ in vitro translation system). Since then, ivt mRNAs are based on this format and contain usually globin 5′ and 3′ UTRs. From the very first report on the possibility to express proteins in animals using injections of synthetic mRNAs [5], as well as early reports of vaccination in mice using synthetic mRNA [6,7], to the initial clinical studies started in 2003 for the evaluation of ivt mRNA vaccines in humans [8,9], all used globin UTR-containing mRNAs. The possibility to replace the initial globin UTRs with better UTRs, providing enhanced functionality, started with the demonstration that a tandem repeat of the 3′ globin UTR brings further stability [10]. Thereafter, academic teams and companies have screened for optimal 5′ and 3′ UTRs using different technologies in order to identify sequences that bestow better functionality to the ivt mRNA of interest in the desired cell type compared to the globin UTRs. As a result, for example, the ivt mRNA contained in the Comirnaty^®^ vaccine [11,12] has a 52 base 5′ UTR from human alpha globin and a 295 base 3′ UTR from two genes: *mtRNR1* and *AES*, while the SpikeVax^®^ vaccine [13,14] has a 57 base 5′ UTR of undisclosed origin and a 111 base 3′ UTR from human alpha globin. That makes a total of 347 nucleotides of UTR for Comirnaty^®^ and 168 nucleotides of UTR for SpikeVax^®^. The secondary and tertiary structures in UTRs allow them to be specifically recognized by proteins, bringing higher stability/translation to the synthetic mRNA. At the same time, these highly complex structures of UTRs can be problematic sequences to introduce in a DNA template and to transcribe. Thus, reducing their length would be an advantage but it was never addressed. In addition, UTRs added to ivt mRNA are supposed to have a specific role (usually bringing stability and enhanced translation) but depending on their precise folding (that may be also affected by the coding sequence, should there be some possible base pairing) and depending on the cell type, they may also be deleterious (recognition by miRNA or other types of RNA present in specific cell types). Thus, we tested whether we could reduce UTRs to a minimal sequence. We previously reported that the 3′ UTR can be totally eliminated by joining the stop codon to the poly-A tail [15]. We now decreased the size of the 5′ UTR to four nucleotides and found that these ivt mRNAs are as functional as usual ivt mRNAs containing hundreds of nucleotides in UTRs. Our “minimal” design greatly facilitates the production of the DNA matrix needed to produce mRNA and could limit potential side effects that longer UTRs could have in cells by interacting with unpredicted RNA or proteins.

## 2. Materials and Methods

### 2.1. Production of Synthetic Messenger RNA by In Vitro Transcription

DNA templates were either PCR fragments or synthetic gene fragments (Twist Bioscience, San Francisco, CA, USA or Azenta, Burlington, MA, USA). They contained a modified T7 promoter accommodating the incorporation of the CleanCap^®^ Reagent AG (TriLink BioTechnologies Inc., San Diego, CA, USA) followed by the desired sequences to be contained in the designed mRNA. Synthetic mRNAs were produced by in vitro transcription using the HiScribe^TM^ T7 mRNA kit (New England Biolabs, Ipswich, MA, USA) at the “ivt mRNA production and formulation platform” in Zurich (https://www.cancer.uzh.ch/en/Research/mRNA-Platform.html, accessed on 22 July 2024). The mRNAs were either co-transcriptionally (PCR fragments in which the reverse primer contained a defined stretch of Ts) or post-transcriptionally (synthetic DNA fragments obtained from companies) poly-adenylated at the 3′ end. The transcription of mRNA was performed in the presence of nucleotides with the four canonical bases (ATP, CTP, GTP, and UTP) or with analogues of uracil instead of UTP: pseudouridine triphosphate, methyl-1 pseudouridine triphosphate, or 5-methoxy uridine triphosphate (TriLink BioTechnologies Inc., San Diego, CA, USA). After transcription, DNA was degraded by DNase I (New England Biolabs, Ipswich, MA, USA). At the same time, if needed (for post-transcriptional polyadenylation), poly-A polymerase (New England Biolabs, Ipswich, MA, USA) and 1 mM ATP were added. Afterward, synthetic mRNA was recovered by LiCl precipitation. RNA was diluted in RNase-free water, and the concentration (after quantification using a Nanodrop—Thermo Scientific, Waltham, MA, USA) was adjusted to 1 mg/mL. The quality and integrity of ivt mRNAs were checked using agarose gel electrophoresis. The mRNAs were stored at −20 °C.

### 2.2. Cells and Transfection

Human embryonic kidney (HEK)-293 cells (from ATCC) and human primary fibroblasts from skin dermis (a generous gift from Dr. Hans-Dietmar Beer) were maintained in RPMI medium (Sigma-Aldrich, St. Louis, MO, USA) containing 10% fetal calf serum (FCS, Gibco^TM^, Waltham, MA, USA) and 0.2% antimicrobial reagent Normocin (InvivoGen, San Diego, CA, USA). The human malignant glioma cell line LN-229 was kindly provided by Dr N. de Tribolet and the human malignant glioma cell line ZH-161 was established from freshly dissected tumor tissue in the Department of Neurology at the University Hospital Zurich, respectively. LN-229 cells were maintained in Dulbecco’s Modified Eagle Medium (DMEM), containing 2 mM L-glutamine, 1% penicillin/streptomycin, and 10% fetal calf serum. ZH-161 cells were maintained in neurobasal medium (Gibco^TM^, Waltham, MA, USA) supplemented with 20 ng/mL fibroblast growth factor-2 and EGF (PeproTech, Rocky Hill, PA, USA), 20 μL/mL B-27 (Gibco^TM^, Waltham, MA, USA) and 2 mM L-glutamine. Human peripheral mononuclear blood cells (PBMCs) were obtained from peripheral blood from healthy donors using the Ficoll-Paque^TM^ Plus (Cytiva, Marlborough, MA, USA) method. Authorization: ECOGEN A220006-02.

Transfection of HEK-293 cells was performed with 100,000 cells per well in 200 μL of RPMI medium supplemented with 10% FCS and 0.2% Normocin using a master mix (53 μL containing 2 μL of Lipofectamine transfection reagent MessengerMAX (Invitrogen^TM^, Waltham, MA, USA) and 1 μg of mRNA in Opti-MEM) and adding 1 μL of it per cell culture well. Transfection of PBMCs was performed with 1 million cells per well in 200 μL of RPMI medium, with supplements, by adding 10 μL of the master mix (equivalent to 200 ng of mRNA). Transfection in those conditions was not inducing any toxicity (Jarzebska et al., Appendix A [16]). Luciferase activity was recorded 24 h after transfection by adding 25 μL of BrightGlo^®^ substrate (Promega, Madison, WI, USA) to the cells (firefly luciferase) or by mixing 20 μL of cell supernatant with 20 μL of Pierce Gaussia Luciferase Glow Assay working solution (Thermo Scientific, Waltham, MA, USA) (gaussia luciferase) and measuring activity using a GloMax^®^ Discover Microplate Reader (Promega, Madison, WI, USA).

For ZsGreen experiments, the transfection of HEK-293 cells was performed with 100,000 cells per well in 200 μL in RPMI medium by adding a mixture of 100 ng of mRNA in 2.5 μL of Opti-MEM and 0.2 μL of Lipofectamine MessengerMAX in 2.5 μL of Opti-MEM to each well. Fluorescence (Excitation: 485 nm; Emission: 528 nm) was recorded in real time over 48 h (after this time, the evaporation of water may affect cell survival) at 37 °C using the Cytation^TM^ 3 Cell Imaging Multi-Mode Reader (BioTek Instruments, Inc., Winooski, VT, USA).

For the expression of SARS-CoV-2 spike, the transfection of HEK-293 cells was performed in a 24-well plate with 1 million cells per well in 1 mL of RPMI medium by adding a mixture of 1000 ng of mRNA in 25 μL of Opti-MEM and 2 μL of MessengerMAX in 25 μL of Opti-MEM to each well. After an incubation of 24 h (at 37 °C, 5% CO_2_ in a humidified atmosphere), cells were harvested by using Trypsin–EDTA (0.25%, Sigma-Aldrich, St. Louis, MO, USA), washed twice with PBS (1×, Sigma-Aldrich, St. Louis, MO, USA), and resuspended with 1 μL Zombie NIR^TM^ (BioLegend, San Diego, CA, USA) in 100 μL of PBS. Cells were incubated for 30 min at 4 °C in the dark. Then, cells were washed twice with PBS and resuspended in 1 μL mouse anti-human SARS-CoV/SARS-CoV-2 (COVID-19) spike antibody [1A9] (GeneTex, Irvine, CA, USA) in 100 μL PBS, followed by an incubation of 45 min at 4 °C in the dark. Again, cells were washed twice and resuspended in 100 μL PBS containing 1 μL PE goat anti-mouse IgG antibody (BioLegend, San Diego, CA, USA). Cells were incubated for 45 min at 4 °C in the dark. The wash steps were repeated and cells were resuspended in 100 μL Paraformaldehyde (1%), and incubated at 4 °C in the dark. A final wash was performed and cells were resuspended in 250 μL PBS and stored at 4 °C in the dark until sample acquisition with BD LSRFortessa^TM^ (BD Biosciences, Franklin Lakes, NJ, USA) and BD FACSDiva^TM^ Software (BD Biosciences, Franklin Lakes, NJ, USA; v8.0.1). The obtained data were analyzed and plotted using FlowJo^TM^ (BD Biosciences, Franklin Lakes, NJ, USA; V 10.0.8) by following the gating strategy: Cells (FSC-A × SSC-A) -> Singlets (FSA-A × FSC-H) -> Live cells (APC-Cy7-A × FSC-A) -> Spike+ cells (PE-A × FSC-A). For the expression of IL-2, 2 ng of mRNA with 0.004 μL of MessengerMax were added to 100 μL of cells (HEK-293 or fibroblasts). After four days of culture, the total amount of IL-2 in the supernatant was measured using ELISA according to the manufacturer’s protocol (ELISA MAX^TM^ Standard Set Mouse IL-2 from BioLegend, San Diego, CA, USA).

### 2.3. CAR T-Cell Generation

Human T cells were isolated from healthy donors (ethic approval 2019-02027 “Multimodal characterization of the central nervous system neoplasms”) using the EasySep^TM^ Release Human CD3 Positive Selection Kit (Stemcell Technologies, #17751, Vancouver, BC, Canada) and activated with CD3/CD28 activation beads (#11131D, Gibco^TM^, Waltham, MA, USA) for 3 days. They were continuously kept in RPMI medium containing 2 mM L-glutamine (Gibco^TM^, Waltham, MA, USA), 1% penicillin/streptomycin, 10% fetal calf serum (FCS, Gibco^TM^, Waltham, MA, USA), and 0.05 mM 2-mercaptoethanol supplemented with 100 U/mL IL-2 and used between Day 11–13 after isolation.

For CAR T-cell generation, 10 μg *NKG2D-CAR-T2A-RQR8* mRNA was electroporated using a NEON transfection system (Invitrogen, Waltham, MA, USA). RQR8 expression was detected with the anti-CD34-PE antibody (Thermo Fisher #MA1-10205, Waltham, MA, USA).

### 2.4. In Vitro Lysis Assay

Tumor lysis assays were performed as previously described [17]. Briefly, 25,000 human glioma cells (LN229 or ZH161) were surface stained with PKH26 (Sigma-Aldrich, St. Louis, MO, USA) and seeded into 96-well plates. CAR T cells were added at different effector:target (E:T) ratios and tumor cell viability was assessed 24 h later using the Zombie Violet Fixable Viability Kit (BioLegend, San Diego, CA, USA) with flow cytometry. Tumor cell lysis was determined as the percentage of death in the population of labeled tumor cells.

### 2.5. Animals and In Vivo Imaging

C57BL/6 mice were kept at the Laboratory Animal Service Center (LASC) facilities in Schlieren (Switzerland) under specific pathogen-free conditions (SPF) with food and water provided ad libitum. All experiments were performed according to the governmental and institutional guidelines and approved by the Veterinary Office of the University of Zurich (Kanton Zurich, Health Direction, Veterinary Office, Zollstrasse 20; 8090 Zurich; license number ZH215/17 and license number ZH004/2021). Animals were purchased from Envigo (Horst, The Netherlands). 

Mice of 4 to 8 weeks of age were injected subcutaneously with 2 μg of mRNA formulated in lipid nanoparticles (LNPs) as previously described [18]. Ten hours later, bioluminescence in vivo imaging was performed on an IVIS Lumina instrument (PerkinElmer, Waltham, MA, USA). Before the measurements, D-luciferin (Synchem UG & Co., KG, Felsberg, Germany) dissolved in PBS (15 mg/mL stock) and sterile-filtered was injected (150 μg/g intraperitoneally). Emitted photons from live animals were quantified 10–20 min post luciferin injections, with an exposure time of 1 min. Regions of interest (ROI) were quantified for average luminescence (counts) (IVIS Living Image 3.2). For vaccination, mice received three subcutaneous injections of 2 μg of mRNA in LNPs. IgG in serum was measured by ELISA (coated with ovalbumin made in-house or coated with SARS-CoV-2 spike, purchased from Krishgen Biosystems, Mumbai, India) four weeks after the last injection. 

## 3. Results

### 3.1. Minimal 5′ UTRs Are Compatible with the Translation of ivt mRNA 

In 2008, Elfakess and Dikstein reported that approximately 4% of mammalian mRNAs possess a very short 5′ UTR sequence that they called TISU (Translation Initiator of Short 5′ UTR) [19]. This sequence is CAAG and is different from the usual Kozak element [19]. We generated mRNAs starting with a Cap1 (CleanCap^®^ m6GpppAG), followed by the CAAGAUG sequence and encoding gaussia luciferase, firefly luciferase, or ZsGreen. These mRNAs have no 3′ UTR (UAA stop codon directly followed by a poly-A tail as previously described [15]). As controls, we made mRNAs with no 5′ UTR or a short Kozak (sequences in Appendix A). Unmodified mRNAs (ACGU) transfected into HEK-293 cells gave in both cases (for gaussia and firefly luciferase) better expression with the TISU 5′ UTR and little expression without a 5′ UTR (Figure 1A). In the case of ZsGreen, considering expression duration as a surrogate for mRNA stability, within 2 days the expression kinetics indicate a better stability of the TISU mRNA (Figure 1B). In order to define the optimal length of the poly-A tail for a minimal ivt mRNA with only four nucleotides in the 5′ UTR, mRNA was generated from DNA constructs containing 0, 20, 40, 60, 80, 100, and 120 As templates. In total, 90 As are sufficient to obtain full expression of ivt mRNA with a total of four bases in the 5′ UTR (Figure 2). To test expression in immune cells that disrupt translation when transfected with ACGU mRNA (through the activation of the anti-viral response and the production of type I interferon), we generated methyl-1 pseudoU-modified mRNAs lacking a 5′ UTR, having a short Kozak, or having the TISU 5′ UTR and tested expression in fresh human PBMCs. As shown in Figure 3, like unmodified mRNAs, methyl-1PseudoU-modified mRNAs encoding gaussia luciferase, firefly luciferase, or ZsGreen are better expressed in tumor cells or immune cells when preceded by a TISU 5′ UTR than when they contain a short Kozak or lack a 5′ UTR (Figure 3A–C). Enriching the coding sequence with AG or CG content (Appendix A) or adding multiple stops (TGA-TAA or TGA-TAG-TAA, Appendix A) at the end of the coding sequence did not increase luciferase expression from mRNAs having a TISU 5′ UTR with no 3′ UTR in HEK-293 cells or PBMCs.

### 3.2. Comparing Minimised ivt mRNA (minRNA) to Classical Optimized ivt mRNAs

As TISU performs best as a short 5′ UTR, we next compared the four-nucleotide ivt UTR mRNA (subsequently referred to as ‘minRNA’) with conventional ivt mRNA sequences with optimized UTRs (optRNA, an eIF4G aptamer as a 5′ UTR [20] and a beta-globin tandem repeat as a 3′ UTR [10]) (Figure 4A). minRNA and optRNA were generated containing different modifications: complete substitution of U residues by pseudoU (ψ), methyl-1 pseudoU (m1ψ), methoxy U (moU), or with no substitution (U). In PBMCs, ψ optimized mRNA coding for gaussia luciferase gave more expression than minRNA (Figure 4B) and in HEK-293 cells moU optimized ivt mRNA coding firefly luciferase showed more protein expression than minRNA (Figure 4C). Despite this, there were no consistent differences in expression between modified minRNA and optimized ivt mRNA types for all three constructs (encoding gaussia luciferase, firefly luciferase, and ZsGreen) in both immortalized cells and PBMCs (Figure 4B–D). Of note, since all experiments are performed with constant amounts of mRNA in terms of nanograms per well, the minRNA formulations contain slightly more RNA molecules than opt mRNA formulations, the former being slightly shorter than the latter.

### 3.3. minRNA for Therapies and Vaccination

After demonstrating, using three reporters (gaussia luciferase, firefly luciferase, and ZsGreen), several mRNA formats (unmodified and modified), and several cell types, that the minRNA design was broadly comparable, in terms of expression of the encoded protein, to the optimized classical ivt mRNA design, next we tested the use of minRNA in the context of relevant vaccines and therapeutics and compared it to the most recent ivt mRNA designs, as contained in Comirnaty^®^ and SpikeVax^®^. For the therapeutic approach, we built on our previous demonstration that CAR T cells engineered with mRNA encoding an NKG2D-CAR could kill glioma cells [17]. We compared the minRNA encoding the NKG2D-CAR and the reporter protein RQR8 in cis [21] with an optimized mRNA having the SpikeVax^®^ 5′ and 3′ UTRs (both methyl-1 pseudoU-modified). The percentage of expression of RQR8 on the surface of electroporated T cells was similar when using the minRNA design and an optimized (based on SpikeVax^®^) design (Figure 5A), although the signal intensity was slightly higher with minRNA. As a result, target cell killing was similar with CAR T cells engineered with minRNA or optimized mRNAs (Figure 5B). Therapies with mRNA coding for IL-2 (eventually mutated or fused to albumin) are promising in the context of cancer [22] and autoimmunity [23]. Thus, we generated minRNA and optimized (5′ and 3′ UTRs from SpikeVax^®^) ivt mRNA (both methyl-1 pseudoU-modified) coding for IL-2, which were transfected into HEK-293 cells and fibroblasts and IL-2 was measured by ELISA after four days of culture. As shown in Figure 5C, the two synthetic mRNAs generated similar amounts of IL-2 from transfected cells (minRNA being slightly better than optimized mRNA in HEK-293 cells but not in primary cells). In vivo, we tested luciferase expression in mice following injection of ivt min or optimized (an eIF4G aptamer as a 5′ UTR [20] and a beta-globin tandem repeat as a 3′ UTR [10]) mRNAs (both methyl-1 pseudoU-modified) formulated in LNPs. Light emission recorded by in vivo imaging indicated similar functionality of the two mRNA types in vivo (Figure 5D,E). We continued with vaccination, comparing the minRNA (methyl-1 pseudoU-modified) encoding ovalbumin or the SARS-CoV-2 spike with the equivalent optRNA having a 5′ UTR consisting of the eIF4G aptamer and the 3′ UTR of Comirnaty^®^ mRNA (*mtRNR1*-*AES*). In transfected cells, the two mRNAs coding for SARS-CoV-2 spike gave equivalent expression of the viral protein on the cell surface (Appendix A). After three subcutaneous injections of the LNP-formulated mRNAs, antibody responses against ovalbumin and spike in mice injected with the mRNA encoding ovalbumin or spike, respectively, were equivalent between the minimized and optimized design (Figure 5F,G).

## 4. Discussion

The UTRs of messenger RNAs have an impact on the rate of translation and the half-life of mRNA in cells [24,25,26]. Historically, ivt mRNA contained the UTRs of globin genes. More recently, better UTRs have been defined as providing an advantage over UTRs from globin genes [25]. However, we re-evaluated the role of UTRs in the context of the current tools available to produce synthetic mRNAs, including a highly efficient cap analog [27] and the ability to make modifications to the mRNA (for example, methyl-1 pseudoU [28]) that affect its functionality. 

We report that synthetic mRNAs with only a total of four UTR nucleotides, specifically a 5′ UTR corresponding to the TISU element and lacking a 3′ UTR, are equally as functional as commonly used ivt mRNAs with optimized 5′ and 3′ UTRs (e.g., in approved vaccines). This holds true regardless of whether the mRNA contains unmodified or modified uracil residues. Our data have a major impact on the future of mRNA-based vaccines and therapies, as our design makes it easier to produce the DNA template required for mRNA production. Indeed, when ordering synthetic genes, structured or repeated UTRs can cause problems and lead to delays and additional costs. When producing the DNA template by PCR, the introduction of the usual UTRs requires very long or overlapping complex PCR primers. A forward primer of 44 nucleotides (T7 promoter followed by 4 TISU nucleotides and 20 complementarity bases) and a reverse primer of 23 nucleotides (complementarity to the targeted gene followed by an extra stop codon; polyadenylation of the synthetic mRNA is then carried out post-transcriptionally using a poly-A polymerase) are all that is needed to generate the template required for in vitro transcription of a functional mRNA from any DNA (including the cell’s total cDNA). Thus, whatever the method chosen, obtaining the DNA template is much easier when producing minRNA than when producing typical optimized ivt mRNAs. In addition, the minRNA design reduces the complexity of the ivt mRNA, thereby reducing the potential unexpected effects of UTRs in different cell types and according to different RNA modifications. Indeed, UTRs validated to enhance the expression of an mRNA sequence in one cell type do not guarantee that other open reading frames (possibly forming dsRNA structures with UTRs) or other cell types (expressing, for example, a microRNA [29] capable of recognizing UTRs and therefore cleaving the synthetic mRNA) will recapitulate these findings. By minimizing the ivt mRNA sequence, the risk of aberrant transcripts and aberrant translations will be reduced (for example, an atypical start in the 5′ UTR could give rise to unexpected proteins [30] or override of the stop codon giving translation of the 3′ UTR [31]).

Now that ivt mRNA has proved to be an advantageous and versatile vaccination tool, whether modified (Spikevax^®^ from Moderna [13] and Comirnaty^®^ from BioNTech/Pfizer [11]) or unmodified (CVcoV^®^ [32,33,34,35], efficacious but not marketed, approved ARCT-154^®^ [36] from Arcturus), numerous vaccines against pathogens, cancers, allergies, and autoimmunity are being developed [37]. At the same time, the immense potential of synthetic mRNA in therapies is beginning to be discovered and is the subject of many clinical studies: for example, for CAR T-cells, regeneration of blood vessels, expression of therapeutic proteins, etc. Thanks to our minRNA design, the production of synthetic mRNAs will be easier, faster, and cheaper, enabling the acceleration of all approaches in which synthetic mRNA is a promising vector for the therapies and vaccines of the future.

## Figures and Tables

**Figure 1 cells-13-01242-f001:**
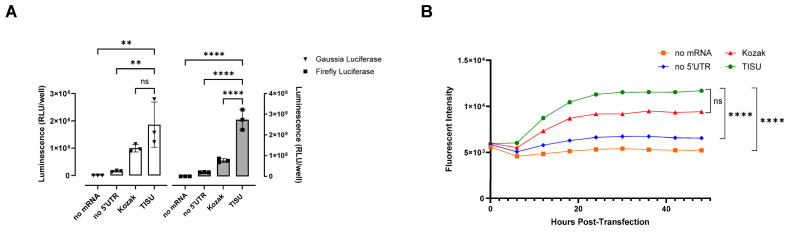
Functionality of the TISU 5′ UTR in unmodified mRNA. (**A**) Luciferase activity in HEK-293 cells 24 h after transfection of unmodified mRNAs containing no 5′UTR, a Kozak 5′UTR, or a TISU 5′ UTR and coding for gaussia luciferase (white bars) or firefly luciferase (grey bars). (**B**) Fluorescence over 48 h in HEK-293 cells after transfection of unmodified mRNAs containing no 5′UTR, a Kozak 5′UTR, or a TISU 5′ UTR and coding for ZsGreen. (**A**,**B**) Experiments were performed in triplicate and data are presented as the mean ± SD. One-way analysis of variance (ANOVA) with Tukey’s multiple comparison test was used to determine the significance between the different groups. ns = not significant, ** *p* < 0.01, **** *p* < 0.0001.

**Figure 2 cells-13-01242-f002:**
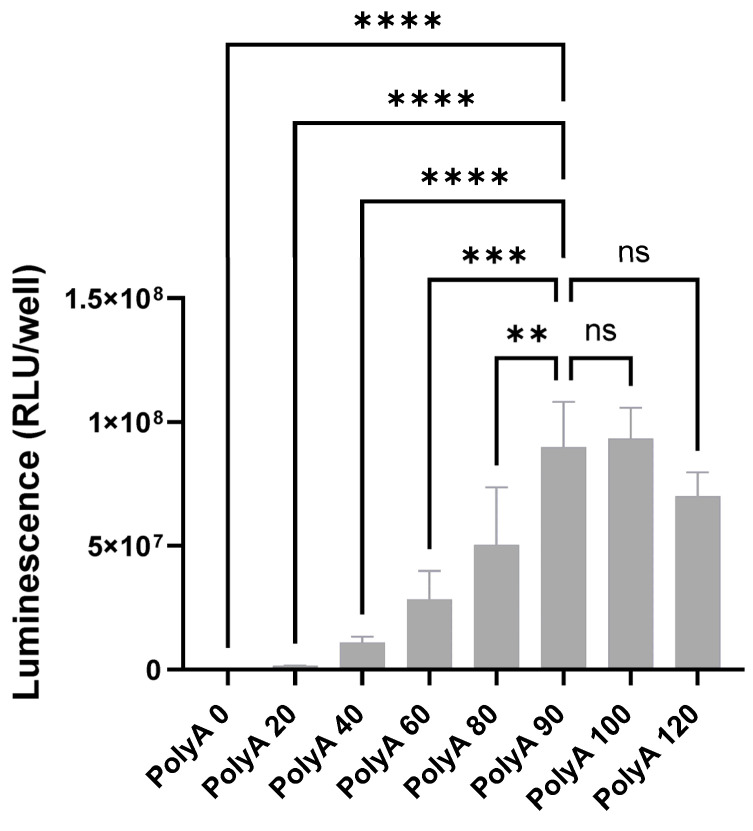
Impact of the length of the poly-A tail on the translation of the TISU 5′ UTR. Luciferase activity in HEK-293 cells 24 h after transfection of mRNAs coding for gaussia luciferase and containing a TISU 5′ UTR and different poly-A tail lengths. The experiment was performed in triplicate and data are presented as the mean ± SD. They were analyzed with one-way ANOVA. ns: not significant; ** *p* < 0.01; *** *p* < 0.001; **** *p* < 0.0001.

**Figure 3 cells-13-01242-f003:**
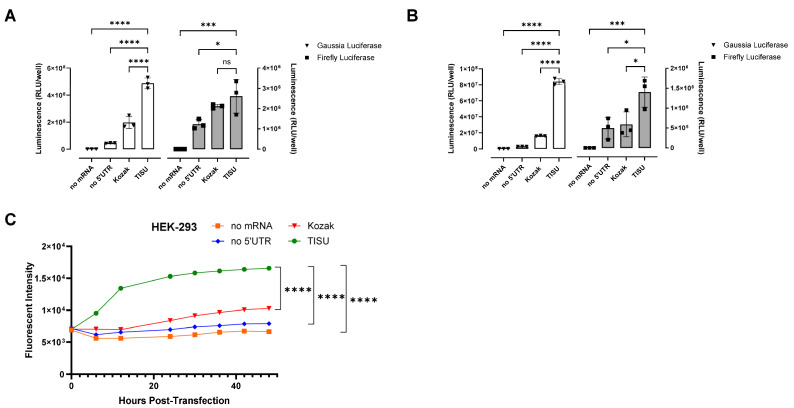
Functionality of the TISU 5′ UTR in methyl-1 pseudoU modified mRNA. (**A**,**B**) Luciferase activity in HEK-293 cells (**A**) and in human blood PBMCs (**B**) 24 h after transfection of methyl-1 PseudoU modified mRNAs containing no 5′UTR, a Kozak 5′UTR, or a TISU 5′ UTR and coding for either gaussia luciferase (white bars) or firefly luciferase (grey bars). (**C**) Fluorescence over 48 h in HEK-293 cells after transfection of methyl-1 PseudoU mRNAs containing no 5′UTR, a Kozak 5′UTR, or a TISU 5′ UTR and coding for ZsGreen. (**A**–**C**) Experiments were performed in triplicate and data are presented as the mean ± SD. One-way ANOVA with Tukey’s multiple comparison test was used to determine the significance between the different groups. ns = not significant, * *p* < 0.05, *** *p* < 0.001, **** *p* < 0.0001.

**Figure 4 cells-13-01242-f004:**
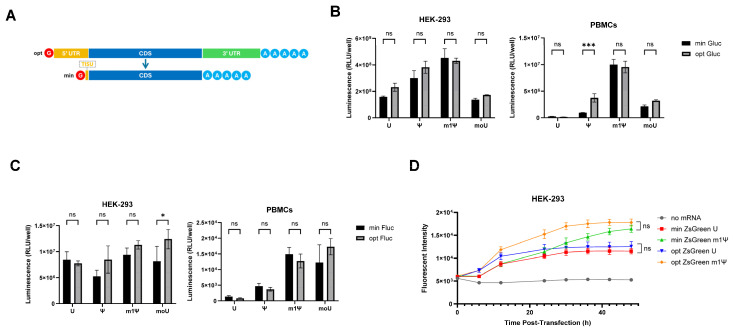
Comparison of minRNA and optimized mRNAs in translating protein. (**A**) Schematic representation of the optimized (“opt”) and minimized (“min”) mRNAs. Depicted are the 5′ and 3′ UTRs of opt mRNA and the coding sequence (CDS) of both. minRNA lacks the conventional 5′ and 3′ UTRs and contains the 4-nucleotide-long TISU Translation Initiator of Short 5′ UTR (TISU) element. (**B**,**C**) Luciferase activity 24 h after transfection of min and opt mRNAs coding for (**B**) gaussia luciferase or (**C**) firefly luciferase in HEK-293 cells (left panels) and PBMCs (right panels). The mRNAs were unmodified or had full replacement of U residues by modified uracils (PseudoU “ψ”, methyl-1 Pseudo U “m1ψ”, or methoxy U “moU”). (**D**) Fluorescence in HEK-293 cells over 48 h after transfection of unmodified or methyl-1 pseudoU modified mRNAs coding for ZsGreen. (**B**,**D**) Experiments were performed in triplicate and data are presented as the mean ± SD. (**B**,**C**) Two-way or (**D**) one-way ANOVA with Tukey’s multiple comparison test was used to determine the significance between the different groups. ns = not significant, * *p* < 0.05, *** *p* < 0.001.

**Figure 5 cells-13-01242-f005:**
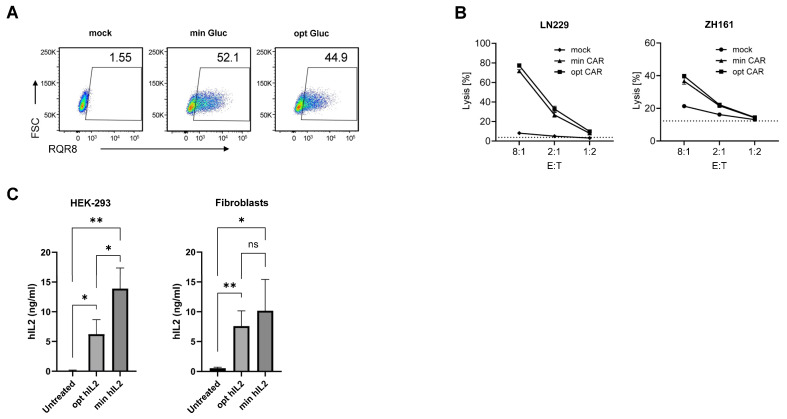
minRNA design coding for therapeutic proteins. (**A**,**B**) Human T cells were mock transfected or transfected with min or opt mRNA coding for NKG2D-CAR-T2A-RQR8. (**A**) Flow cytometry analysis for RQR8 surface expression 24 h after transfection. RQR8-positive cells were detected using anti-CD34-PE antibody (RQR8; PE: phycoerythrin) and forward scatter (FSC). (**B**) Flow cytometry quantifications of glioma cells surface-stained with PKH26 and incubated for 24 h with human T cells at different E:T ratios. Tumor cell viability was assessed using Zombie Violet and cell lysis was determined as the percentage of death in the population of labeled tumor cells. Unspecific background lysis is represented by dotted lines. (**C**) minRNA and optimized mRNA coding human IL-2 were transfected into HEK-293 or human primary fibroblasts. Supernatants were measured 96 h later for IL-2 production by ELISA. Experiments were performed in triplicate and data presented as the mean ± SD. An unpaired, two-tailed Student’s *t*-test was used to determine the significance between the different groups. ns = not significant, * *p* < 0.05, ** *p* <0.01. (**D**) Representative IVIS images of mice untreated or injected subcutaneously with mRNA coding for firefly luciferase with either the opt or min mRNA design. Regions of interest were quantified for average luminescence (counts) (IVIS Living Image 4.0) (**E**) Quantification of the photon flux shown in (**D**). (**F**,**G**) ELISA quantification of serum antibodies against ovalbumin or spike in mice injected subcutaneously with LNP containing min or opt mRNA coding ovalbumin or spike. (**E**–**G**) N = 5 mice per group. Data are presented as the mean ± SD. (**E**) One-way ANOVA with Tukey’s multiple comparison test or (**G**) an unpaired, two-tailed Student’s *t*-test was used to determine the significance between the different groups. ns = not significant, ** *p* < 0.01.

## Data Availability

Data associated with this paper can all be found within the article and Appendix A.

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
