# Peer review of "Synthetic mRNAs Containing Minimalistic Untranslated Regions Are Highly Functional In Vitro and In Vivo"

_cells, 2024, doi:10.3390/cells13151242_

Round 1
Reviewer 1 Report
Comments and Suggestions for Authors
Comments to authors:
The manuscript of an article, which was written by Dr. Shahab Mamaghani et al, is interesting, examining if shorter RNAs could be replaced with the presently used UTR-containing RNAs and if they can be applied on the vaccination. However, I would suggest authors to edit text and Figures. Authors had better revise each Figure legends just providing essential minimal information but eliminate the details on the text. More importantly, Figures should be as concise as possible, without wide blanks. The units for each vertical axis of all Figures should be reconsidered.
Recommendation: Major revision
General comments
Experimental design and the performance are good. However, in this manuscript, authors had better provide required but minimum information on each Figure legends. Details of the experimental conditions and methods are not usually written in the text but Figure legends. Additionally, Figures should be redrawn to make them as concise as possible without losing essential points.
Specific comments
P3, L98: Origin of the HEK-293 cells should be provided with appropriate citations.
P3, L102: If human PBMCs were used for experiments, it would better be done in accordance with some permissions to obey the law.
P3, L129: CO2, 2 should be subscript.
P4, L190: References are needed (Elkafes & Dikstein)
For all Figures: Luciferase activity should be normalized by time for counting RLU, and protein amount or cell number. Therefore, the vertical axis should be “RLU/ng/sec”, “RLU/sec/10000 cells” or the other. Readers will wonder the experimental condition for the Luciferase assay. Similarly, Fluorescent Intensity should be normalized.
P4, Figure 1: Delete “HEK-293” from each panel. (A) and (B) can be combined. For example, how about “gaussia” and “firefly” are indicated in red and green bars, respectively, and the red and green vertical axis can be added on both sides? Anyway, wide blank area on the center should be avoided.
P7, Figure 3A and 3B: Similar to the suggestion for the Figure 1, the gaussia and firefly Luciferase activities can be combined within the results from HEK-293 or that from PBMC. That will greatly reduce the unprinted area.
P8, Figure 4A: Information for the UTRs and CDS should be provided in the legend.
P8, L256: “48h”, “48 hours”. Unify the units all through the text.
P8, L260: “*p<.05, ***p<.001”, “p<0.05, ***p<0.001”. Unify the indication all through the text.
P9, Figure 5A: Explanations for FSC-A and PE should be written in the legend.
P9, Figure 5 B: How or by what assay system, Lysis (%) was measured and calculated? That should be briefly commented in the legend.
P9, Figure 5C: Detailed ELISA conditions that were not written in “Materials and Methods” should be written in the legend. Additionally, readers will wonder what kind of fibroblasts were used.
P10, Figure 5D: Aren’t units required for the color scale?
P10, Figure 5E-G: Delete “LNP Vaccination”, “OVA Vaccination” and “Spike Vaccination” from each panel. The information had better be indicated in the legend not on the Figure.
P10, Figure 5F: Absorbance (AU) is obscure. At least wavelength should be indicated. This panel can be made into compact if four bars are indicated as groups.
P10, Figure 5: The min RNAs did improve translation of IL2 in vivo (Fig. 5A-C). However, the min RNAs did not so much improved vaccination efficiencies in vivo comparing the effect of the optRNAs (Fig. 5D-G). Authors had better discuss the differences.
P10, L311-313: “untranslated regions”, “UTRs”. It should be unified all through the text.
P10-11: Were there no studies to reduce nucleotides of the RNAs for vaccination ever? Is the concept for designing short RNAs firstly shown in this article? Moreover, readers will wonder why the RNA length should be shortened. Further discussion would be needed with appropriate citations.
Author Response
General comments
Experimental design and the performance are good. However, in this manuscript, authors had better provide required but minimum information on each Figure legends. Details of the experimental conditions and methods are not usually written in the text but Figure legends. Additionally, Figures should be redrawn to make them as concise as possible without losing essential points.
Response: We thank the reviewer for their comprehensive review of our manuscript. We have made the changes that the reviewer has suggested and given a point-by-point response below. We think the manuscript has greatly improved with the reviewer’s input.
Specific comments
P3, L98: Origin of the HEK-293 cells should be provided with appropriate citations.
Response: HEK-293 cells were obtained from the American Type Culture Collection (ATCC), now changed in the text.
P3, L102: If human PBMCs were used for experiments, it would better be done in accordance with some permissions to obey the law.
Response: We are authorized to use blood from blood donors, our ECOGEN number is A220006-02. Added line 113.
P3, L129: CO2, 2 should be subscript.
Response: It is now corrected in the manuscript.
P4, L190: References are needed (Elkafes & Dikstein)
Response: Thank you. The reference was added line 205.
For all Figures: Luciferase activity should be normalized by time for counting RLU, and protein amount or cell number. Therefore, the vertical axis should be “RLU/ng/sec”, “RLU/sec/10000 cells” or the other. Readers will wonder the experimental condition for the Luciferase assay. Similarly, Fluorescent Intensity should be normalized.
Response: All wells that are compared contain the same number of cells. We suggest to keep our representation as it is the one used even in MDPI article for example: https://www.mdpi.com/1422-0067/24/19/14854
P4, Figure 1: Delete “HEK-293” from each panel. (A) and (B) can be combined. For example, how about “gaussia” and “firefly” are indicated in red and green bars, respectively, and the red and green vertical axis can be added on both sides? Anyway, wide blank area on the center should be avoided.
Response: the figure 1 was accordingly modified.
P7, Figure 3A and 3B: Similar to the suggestion for the Figure 1, the gaussia and firefly Luciferase activities can be combined within the results from HEK-293 or that from PBMC. That will greatly reduce the unprinted area.
Response: the figure 3 was accordingly modified.
P8, Figure 4A: Information for the UTRs and CDS should be provided in the legend.
Response: We have now provided more information on the UTRs, the TISU and CDS in the figure legend of 4A.
P8, L256: “48h”, “48 hours”. Unify the units all through the text.
Response: 24h and 48h are corrected to 24 hours and 48 hours, respectively, throughout the manuscript.
P8, L260: “*p<.05, ***p<.001”, “p<0.05, ***p<0.001”. Unify the indication all through the text.
Response: Thank you. We have now unified this to match the other figure legends.
P9, Figure 5A: Explanations for FSC-A and PE should be written in the legend.
Response: We have now explained this in the figure legend as follows: “RQR8-positive cells were detected using with the anti-CD34-PE antibody (RQR8) and Forward scatter (FSC-A).”
P9, Figure 5 B: How or by what assay system, Lysis (%) was measured and calculated? That should be briefly commented in the legend.
Response: We have now explained this in the figure legend as follows: “Flow cytometry quantifications of glioma cells surface-stained with PKH26 and incubated for 24 hours with human T cells at different E:T ratios. Tumor cell viability was assessed using the Zombie Violet and cell lysis was determined as the percentage of death in the population of labeled tumor cells.”
P9, Figure 5C: Detailed ELISA conditions that were not written in “Materials and Methods” should be written in the legend. Additionally, readers will wonder what kind of fibroblasts were used.
Response: We have now explained this in the figure legend as follows: “minRNA and optimised mRNA coding human IL-2 were transfected into HEK-293 or human primary fibroblasts. Supernatants were measured 96 hours later for IL-2 production by ELISA.”
Fibroblasts are human primary fibroblasts from dermal skin, so we have stated “human primary fibroblasts from skin dermis” on line 100 in Materials & Methods and “ human primary fibroblasts” in Figure legend 5.
P10, Figure 5D: Aren’t units required for the color scale?
Response: Thank you. We have now added the units to the figure legend and the figure 5D.
P10, Figure 5E-G: Delete “LNP Vaccination”, “OVA Vaccination” and “Spike Vaccination” from each panel. The information had better be indicated in the legend not on the Figure.
Response: Thank you. We have now deleted the titles in the graphs 5E-G and additional indications have been added in the legend
P10, Figure 5F: Absorbance (AU) is obscure. At least wavelength should be indicated. This panel can be made into compact if four bars are indicated as groups.
Response: Thank you. We indicated the wavelength and replaced the bar representation by lines.
P10, Figure 5: The min RNAs did improve translation of IL2 in vivo (Fig. 5A-C). However, the min RNAs did not so much improved vaccination efficiencies in vivo comparing the effect of the optRNAs (Fig. 5D-G). Authors had better discuss the differences.
Response: We did not test IL2 in vivo. Figure 5 presents expression of IL-2 in vitro. It is not significantly different between opt and min RNA in primary cells (fibroblasts). We added line 312 “(minRNA being slightly better than optimised mRNA in HEK-293 cells but not in primary cells).”
P10, L311-313: “untranslated regions”, “UTRs”. It should be unified all through the text.
Response: Thank you. We have now unified this throughout the text.
P10-11: Were there no studies to reduce nucleotides of the RNAs for vaccination ever? Is the concept for designing short RNAs firstly shown in this article? Moreover, readers will wonder why the RNA length should be shortened. Further discussion would be needed with appropriate citations.
Response: The concept of reducing the length oof UTRs was never addressed and is firstly shown in this article. We added “Thus, reducing their length would be an advantage but it was never addressed.” Line 63.
Reviewer 2 Report
Comments and Suggestions for Authors
Introduction.
The references cited are appropriate. However, considering the number of reference articles, which is 14 in total, more recent findings could be added to illustrate the recent developments in the mRNA therapy field. For instance, recent approaches by other groups in minimizing and designing UTR could be introduced in the Introduction section. (major comment 1)
Methods.
- A paragraph spanning lines 113 to 147 is too lengthy. To enhance its readability, some improvements that could be made. Repeated usage of terms such as "animicrobial reagent" or "lipofectamine transfection reagent" could be avoided. The company name and location of materials could be mentioned only when they first appear.
- Regarding lines 107 and 145, how was liquid volume below 0.1 microliter quantified? In line 145, the exact number of cells should be clarified.
Results.
- What is the authors' view on the applicability of the minRNA combination on general mRNAs? (major comment 2)
- Regarding Figures 1C, 3C, and 4D, the authors should analyze or discussion the duration of the fluorescent intensity for time points longer than 48 hours. How long do the signals appear? (major comment 3)
- For coherence throughout the manuscript, the emerging order of appearance of Figures 1A and 1B could be switched.
- Regarding lines 246 to 249, the result shows statistical significance in some comparisons. Therefore, the authors' description of "no difference in expression" between mRNA types should be modified.
- For Figures 4 and 5, where the translation efficiency of mRNAs with different nucleotide length was compared, this reviewer assumes that the same nanogram amount of mRNAs were delivered. In this case, their molecule number must be different. Could this affect the comparison result of protein expressions? (major comment 4)
- Regarding the flow cytometry-based results, how was signal intensity different, on top of the positivity (Fig. 5A)?
Comments on the Quality of English LanguagePlease use abbreviations only when they first appear.
More typos in Results: minimal (line 187), HEK-293 (line 279 and Figure 4D), microRNA (line 339), 100,000 (lines 104, 118)
Spacing: Lines 100, 103, 104, 332
Author Response
Introduction.
The references cited are appropriate. However, considering the number of reference articles, which is 14 in total, more recent findings could be added to illustrate the recent developments in the mRNA therapy field. For instance, recent approaches by other groups in minimizing and designing UTR could be introduced in the Introduction section. (major comment 1)
Response: No research aimed at minimizing UTR, this is now mentioned line “63 in the introduction “Thus, reducing their length would be an advantage but it was never addressed”. More references were added (now 21)
Methods.
- A paragraph spanning lines 113 to 147 is too lengthy. To enhance its readability, some improvements that could be made. Repeated usage of terms such as "animicrobial reagent" or "lipofectamine transfection reagent" could be avoided. The company name and location of materials could be mentioned only when they first appear.
Response: We thank the author for pointing this out. We have shortened the text and removed the excessive description of RPMI-supplemented medium. We’ve also looked through the manuscript to correct the number of times a company has been cited. We have also changed the description of Normocin and Lipofectamine to only appear the first time.
- Regarding lines 107 and 145, how was liquid volume below 0.1 microliter quantified? In line 145, the exact number of cells should be clarified.
Response: We do not pipette such small volumes but do a master mix of RNA with Lipofectamine MessengerMax and from it we take the amounts that were written. Now things are clarified lines 117-125.
Results.
- What is the authors' view on the applicability of the minRNA combination on general mRNAs? (major comment 2)
Response: We do not understand this question.
- Regarding Figures 1C, 3C, and 4D, the authors should analyze or discussion the duration of the fluorescent intensity for time points longer than 48 hours. How long do the signals appear? (major comment 3)
Response: Cytation has no humid chamber so sample cannot be incubated over 48 hours (wells are drying). We added line 138 “over 48 hours (after this time evaporation of water may affect cell survival)”.
- For coherence throughout the manuscript, the emerging order of appearance of Figures 1A and 1B could be switched.
Response: The Figures have been re-adjusted to improve coherence in the manuscript and text accordingly adjusted. Figures 1A and 1B have been merged into a single graph 1A.
- Regarding lines 246 to 249, the result shows statistical significance in some comparisons. Therefore, the authors' description of "no difference in expression" between mRNA types should be modified.
Response: We thank the author for this comment. We have changed the text to read: “In HEK-292 cells, moU optimised ivt mRNA coding firefly luciferase showed more signif-icant protein expression than minRNA (Figure 4B) and in PBMCs ψ opt mRNA coding gaussia luciferase showed more significance than minRNA (Figure 4C). Despite this, there was were no consistent differences in expression between modified minRNA and optimised ivt mRNA…..”. Line 258.
- For Figures 4 and 5, where the translation efficiency of mRNAs with different nucleotide length was compared, this reviewer assumes that the same nanogram amount of mRNAs were delivered. In this case, their molecule number must be different. Could this affect the comparison result of protein expressions? (major comment 4)
Response: yes, minRNA being slightly shorter than opt mRNA, it contains more molecules per ng. We added lines 273-276 “Of note, since all experiments are performed with constant amounts of mRNA in terms of nanograms per well, minRNA formulations contain slightly more RNA molecules than opt mRNA formulations, the former being slightly shorter than the latter.”
- Regarding the flow cytometry-based results, how was signal intensity different, on top of the positivity (Fig. 5A)?
Response: Plotted signal intensity gives the following: (graphs can not be paste here, please see the response letter)
We added line 300 “although signal intensity was slightly higher with minRNA”.
Comments on the Quality of English Language
Please use abbreviations only when they first appear.
Response: We’ve looked through the manuscript to correct this, we have changed this for ANOVA, and UTRs (as described above). We think we have caught all multiple abbreviations.
More typos in Results: minimal (line 187), HEK-293 (line 279 and Figure 4D), microRNA (line 339), 100,000 (lines 104, 118). Also need to correct HEK to HEK-293 in Fig 4D.
Response: Thank you. We have now corrected these in the text.
Spacing: Lines 100, 103, 104, 332 –
Response: We couldn’t see this issue now, it must have been corrected in the manuscript format.
Round 2
Reviewer 1 Report
Comments and Suggestions for Authors
Comments to authors:
The manuscript, which was written by Dr. Shahab Mamaghani et al has been almost successfully revised according to the suggestions that I have commented. Importantly, authors have edited Figures with minimal but essential information with appropriate description in each Figure legends. In addition, I evaluate that authors have successfully summarized the text as concise as it could be. However, still the RLU on the vertical axis of each Figure was not normalized. So, I would just suggest authors to check the Units on the y-axis of each Figure again. The value should be normalized by protein amount or cell number. Even though cells were not damaged by transfection at all, it needs normalization. In that case, it should be indicated as RLU/well. If not for the description that cells did not die or get damaged at all, readers will wonder each data if it could be compared with each other. Additionally, authors had better check all through the text again to confirm that it is completely freed from any errors, including typos and formatting of the references.
Recommendation: Accept after minor revision
Author Response
Round 2:
The cells did not die or get damaged at all by the lipofection method used here (mRNA in Lipofectamine MessengerMax) as we have shown (supplementary figure S1 B, Jarzebska et al, now reference 16). So we added lines 126-127 "Transfection in these conditions does not induce any toxicity 16 (Supplementary Figure S1 B)." and changed the y axes in all relevant (Luciferase assays) figures from "Luminescence (RLU)" to "Luminescence (RLU/well)". We hope that this answers the question of the reviewer.